# Autocrine Bradykinin Release Promotes Ischemic Preconditioning-Induced Cytoprotection in Bovine Aortic Endothelial Cells

**DOI:** 10.3390/ijms21082965

**Published:** 2020-04-23

**Authors:** Alessandro Bellis, Daniela Sorriento, Antonella Fiordelisi, Raffaele Izzo, Junichi Sadoshima, Ciro Mauro, Federica Cerasuolo, Costantino Mancusi, Emanuele Barbato, Emanuele Pilato, Bruno Trimarco, Carmine Morisco

**Affiliations:** 1Dipartimento di Scienze Biomediche Avanzate, Università FEDERICO II, via S. Pansini n. 5, Napoli, 80131 Naples, Italy; abellis82@vodafone.it (A.B.); daniela.sorriento@unina.it (D.S.); antonella.fiordelisi@gmail.com (A.F.); rafizzo@unina.it (R.I.); f.andrea_cerasuolo@hotmail.it (F.C.); costantino.mancusi@unina.it (C.M.); emanuele.barbato@unina.it (E.B.); trimarco@unina.it (B.T.); 2Dipartimento Emergenza Accettazione, Unità Operativa Complessa Cardiologia con UTIC ed Emodinamica, Azienda Ospedaliera “Antonio Cardarelli”, via A. Cardarelli n. 9, Napoli, 80131 Naples, Italy; ciro.mauro1957@gmail.com; 3Department of Cell Biology and Molecular Medicine, Rutgers–New Jersey Medical School, 185 S. Orange Ave, MSB G609, Newark, NJ 07103, USA; sadoshju@njms.rutgers.edu; 4Dipartimento di Emergenze Cardiovascolari, Medicina Clinica e dell’Invecchiamento, Azienda Ospedaliera Università, FEDERICO II, via S. Pansini n. 5, Napoli, 80131 Naples, Italy; emapilato@yahoo.it

**Keywords:** inflammation, bradykinin, hypoxia, cell protection, Akt

## Abstract

The aims of this study were to assess whether ischemic preconditioning (PC) induces bradykinin (Bk) synthesis in bovine aortic endothelial cells (bAECs) and, if so, to explore the molecular mechanisms by which this peptide provides cytoprotection against hypoxia. PC was induced by exposing bAECs to three cycles of 15 min of hypoxia followed by 15 min of reoxygenation. Bk synthesis peaked in correspondence to the early and late phases of PC (10^−12^ M and 10^−11^ M, respectively) and was abolished by a selective tissue kallikrein inhibitor, aprotinin. Stimulation with exogenous Bk at concentrations of 10^−12^ M and 10^−11^ M reduced the cell death induced by 12 h of hypoxia by 50%. Pretreatment with HOE−140, a Bk receptor 2 (BKR2) inhibitor, in bAECs exposed to 12 h of hypoxia, abrogated the cytoprotective effect of early and late PC, whereas des-Arg-HOE-140, a Bk receptor 1 (BKR1) inhibitor, affected only the late PC. In addition, we found that PC evoked endocytosis and the recycling of BKR2 during both the early and late phases, and that inhibition of these pathways affected PC-mediated cytoprotection. Finally, we evaluated the activation of PKA and Akt in the presence or absence of BKR2 inhibitor. HOE-140 abrogated PKA and Akt activation during both early and late PC. Consistently, BKR2 inhibition abolished cross-talk between PKA and Akt in PC. In bAECs, Bk-synthesis evoked by PC mediates the protection against both apoptotic and necrotic hypoxia-induced cell death in an autocrine manner, by both BKR2- and BKR1-dependent mechanisms.

## 1. Introduction

A consistent body of evidence [1,2,3,4] has shown that bradykinin (Bk) plays a pivotal role as a systemic mediator of ischemic preconditioning (PC). This is a physiological phenomenon in which non-sustained, repetitive, sub-lethal ischemic stimulation enhances tolerance to subsequent prolonged ischemic stress [5]. Two windows of cytoprotection (early and late PC) have been reported [6]. Bk is a peptide cleaved from kininogen precursors in the interstitium and catabolized by the angiotensin-converting enzyme in the vasculature and by neutral endopeptidases in the interstitium [7,8]. Furthermore, it has been demonstrated that Bk receptor 2 (BKR2) blockade abrogates protection by remote PC [3,9,10,11], allowing speculation that the endothelium represents an important source of Bk release evoked by PC.

We have previously demonstrated that, in bovine aortic endothelial cells (bAECs) exposed to prolonged hypoxia, PC-induced cell protection is a Protein Kinase A (PKA)/Akt-dependent phenomenon and that reactive oxygen species (ROS) are primarily involved in this pathway [12]. In particular, low doses of ROS, by reducing thiolic residues in transmembrane receptors, induce the activation of downstream protection pathways, whereas high doses of ROS, by affecting the tertiary structure of these receptors, inhibit the activation of protective mechanisms [13,14]. Furthermore, it has been reported that endothelial cells are able to synthesize and release Bk through the activity of an endothelial isoform of tissue kallikrein that cleaves endogenous kininogen [15,16]. However, it has not yet been clarified whether endothelium produces Bk during PC. If so, it would be reasonable to hypothesize that the autocrine release of Bk exerts the cytoprotective effect through a PKA/Akt-dependent mechanism.

Thus, using preconditioned bAECs exposed to prolonged hypoxic stress, we investigated (a) whether PC induces Bk production and (b) the molecular mechanisms by which this peptide is synthesized and mediates cytoprotection.

## 2. Results

### 2.1. PC Evokes Bradykinin Release from Endothelial Cells

Firstly, we evaluated whether Bk is synthesized and released by preconditioned bAECs. For this purpose, we measured Bk concentrations in culture medium from plates containing cells exposed to the PC stimulus for the indicated durations. PC was induced in bAECs, as previously described [12]. The Bk concentration was increased at both the early (right after PC, 12.5 ± 3.2 pg/mL, 10^−12^ M) and the late phase of PC (24 h after PC, 161.7 ± 10.2 pg/mL, 10^−11^ M) (Figure 1A). Then, we analyzed the mechanisms by which PC increases Bk synthesis in bAECs. In order to address this question, we evaluated the expression of kallikrein (KLK1), the enzyme selectively involved in Bk synthesis in endothelial cells, and of its substrate Kn by RT-PCR. PC did not increase the mRNA levels of either Kn or KLK1 (Figure 1B,C); thus, Bk synthesis in the two phases of PC cannot be attributed to an augmented expression of these proteins. In fact, KLK1 protein expression does not change during PC (Appendix A). Subsequently, we asked whether an increase in the enzymatic activity of KLK1 could be responsible for the increased Bk release during PC. We assessed KLK1 activity through the colorimetric measurement of the release of 4-nitroaniline from preconditioned bAECs. KLK1 activity was enhanced at both early (2.7 fold vs. control, *p* < 0.001) and late PC (3.5 fold vs. control, *p* < 0.001), suggesting that the increased catalytic activity of this enzyme evokes Bk synthesis during PC (Figure 1D). Consistently, the pretreatment of bAECs with a selective inhibitor of KLK1, AP, abrogated Bk release in both phases of PC (Figure 1E).

These results show that bAECs synthesize Bk during early and late PC through an increase in the activity of KLK1.

### 2.2. PC-Induced Bk Synthesis Promotes Cytoprotection against Hypoxia-Induced Apoptosis

Since Bk is believed to be a key mediator of PC-induced cytoprotection in different experimental settings, we evaluated whether the Bk released during PC can prevent apoptosis in bAECs. For this purpose, we assessed cell death in early and late preconditioned bAECs exposed to prolonged hypoxia in the presence or absence of aprotinin (AP). AP pretreatment abrogated the PC-induced cytoprotective effect; in particular, apoptotic cell death was increased in AP-pretreated early and late preconditioned cells (46% ± 3% and 49% ± 4%, respectively) compared to in non-pretreated early (25% ± 5%) and late (28% ± 4%) preconditioned cells (Figure 2) (Appendix A). Consistently, the stimulation of bAECs with concentrations of exogenous Bk comparable to those found in culture media from early and late preconditioned cells (10^−12^ M and 10^−11^ M) decreased apoptotic cell death (27% ± 5% and 26% ± 2%, respectively) compared to in non-preconditioned cells (48% ± 5%) (Figure 2) (Appendix A). Apoptosis was further explored by analysis of caspase−3 cleavage, which plays a key role in regulation of the cellular suicide cascade [17]. This analysis confirmed PC- and Bk-induced cytoprotection against apoptosis (Appendix A).

These data indicate that, in endothelial cells, PC-mediated Bk release through an autocrine mechanism accounts for the cytoprotection against prolonged hypoxia-induced apoptosis.

### 2.3. PC Induces Cytoprotection through BKRs

Since Bk plays a key role in the PC-induced cytoprotection of bAECs, we asked whether BKRs 1 and 2 are also involved. For this purpose, we assessed cell death in early and late preconditioned bAECs exposed to prolonged hypoxia in the presence or absence of selective BKR inhibitors. Cell pretreatment with HOE-140 (a selective BKR2 inhibitor) abrogated PC-mediated cytoprotection in both PC phases; in contrast, pretreatment with desArg10-HOE-140 (a selective BKR1 inhibitor) only affected late PC (Figure 3A) (Appendix A). Apoptosis was also explored by the analysis of caspase-3 cleavage that increased in both PC phases for HOE-140-pretreated cells (Appendix A), whereas it only did in late PC for desArg10-HOE140-pretreated cells (Appendix A). These results were confirmed by silencing BKR expression through the transient transfection of bAECs with specific siRNAs (Appendix A). These data suggest a primary role of BKR2 in both early and late PC. Consistent with the literature, we found that BKR2 is constitutively expressed in endothelial cells and does not change in expression, as assessed by RT-PCR and Western blotting, during PC (Figure 3B,D). By contrast, BKR1 is not constitutively expressed, and the mRNA levels of BKR1 were increased at both phases of PC (Figure 3C). Furthermore, it is noteworthy that the protein expression of BKR1 was significantly higher during late PC than in early PC (Figure 3E).

These results suggest that BKR2 plays a predominant role in PC-mediated cytoprotection, which is subsequently supported by the increased expression of BKR1 in late PC.

### 2.4. BKR2 Endocytosis and Recycling Induce Cytoprotection During PC

Since we demonstrated that BKR2 plays a primary role in both phases of PC, we analyzed the mechanisms by which this receptor induces cytoprotection. In particular, because it is well known that the sustained stimulation of BKR2 results in the rapid phosphorylation/dephosphorylation of this receptor, followed by its desensitization and internalization [18], we evaluated the role of BKR2 endocytosis during PC. We found that both phases of PC induced BKR2 endocytosis. In particular, during early PC, BKR2 expression on the cell surface was decreased (0.54 fold versus control; Figure 4A) whereas its expression in the cytosol was increased (2.2 fold versus control; Figure 4B). In late PC, BKR2 expression on the cell surface was further decreased (0.2 fold versus control; Figure 4A) and was proportionally increased in cytosolic extracts (2.6 fold versus control; Figure 4B). Consistently, the inhibition of clathrin (the protein that regulates the endocytosis of BKR2) [19] with monodansylcadaverine (10^−4^ M) or of vacuolar ATPase (the proton Na^+^/H^+^ pump that acidifies endosomes, lysosomes and other organelles regulating the recycling of BKR2) [20,21] with bafilomycin (10^−7^ M) negatively affected PC-mediated mechanisms of cytoprotection (Figure 4C) (Appendix A). Apoptosis was further explored by the analysis of caspase−3 cleavage that increased in two PC phases for both monodansylcadaverine- and bafilomycin-pretreated cells (Appendix A).

These findings indicate that the recycling and endocytosis of BKR2 are required for the PC-induced cytoprotective effect.

### 2.5. BKRs Mediate PKA and Akt Activation during PC

Since we previously demonstrated that PC can prevent apoptotic cell death in bAECs through the PKA-dependent activation of Akt [12], we tested whether the stimulation of BKRs during PC induces PKA and Akt activation. We evaluated PKA and Akt activation in early and late preconditioned bAECs in the presence or absence of selective BKR inhibitors. The phosphorylation of Kp, a synthetic substrate of PKA, was assayed as a measure of PKA activation; Akt activation was measured by analysis of phosphorylation at the serine473 residue. The pretreatment of bAECs with HOE-140 abrogated Kp phosphorylation in both phases of PC, whereas the administration of desArg10-HOE-140 inhibited Kp phosphorylation only during the late PC (Figure 5A). Consistently, Akt activation was abolished only in the late PC after pretreatment of bAECs with desArg10-HOE-140 (Figure 5B), whereas HOE-140 abrogated Akt phosphorylation in both phases of PC (Figure 5C).

Because Akt is a serine/threonine kinase that regulates several PKA-[12] and phosphatidil-inositol-3-kinase (PI3K)-dependent biological activities, including cell survival [22], we sought to confirm the inhibition of this kinase by the transient transfection of bAECs with specific siRNAs for BKRs. As with pretreatment with selective receptor antagonists, BKR2 silencing abolished Akt phosphorylation in both phases of PC (Figure 5D), whereas BKR1 silencing only inhibited Akt phosphorylation in late PC (Figure 5E). Furthermore, the pretreatment of bAECs with the selective KLK1 inhibitor AP completely abrogated PC-induced Akt activation (Appendix A). 

Together, these experiments indicate that, during PC, BKR2 plays a pivotal role in PKA and Akt activation.

### 2.6. BKRs Induce Crosstalk between PKA and Akt during PC

Finally, in order to demonstrate that, during PC, Bk induces crosstalk between PKA and Akt through BKR activation, we evaluated whether the pretreatment of bAECs with AP, desArg10-HOE-140 or HOE-140 abrogates the interaction between these two kinases. For this purpose, lysates from cells subjected to PC in the presence or absence of AP and BKR inhibitors were immunoprecipitated with antibodies against the PKA catalytic subunit and then blotted with anti-phospho-Akt (ser473) antibody. AP and HOE-140 abrogated crosstalk between PKA and phospho-Akt in both phases of PC (Figure 6A,B), whereas desArg10-HOE-140 only abolished this interaction in the late PC (Figure 6B).

Together, these experiments indicate that, during PC, Bk release and BKR activation are required for the crosstalk between PKA and Akt.

## 3. Discussion

The main result of this study is that, in bAECs, PC induces Bk synthesis that, through an autocrine mechanism, promotes cytoprotection against prolonged hypoxia. This phenomenon is mainly mediated by the internalization of BKR2, which in turn, promotes the crosstalk between PKA and Akt.

Bk is a peptide that acts through a G-protein coupled receptor (GPCR) and is one of the most important mediators of PC [23]. It is well known that receptors belonging to the GPCR family trigger PC [24]. In this regard, it has been demonstrated that adenosine and opioid receptors mediate PC-induced cytoprotection in the heart. In particular, adenosine-induced cardioprotective effects are mainly mediated by adenosine A1 receptors, whose activation has been shown to protect against ischemia/reperfusion injury and to counteract several processes associated with heart failure, including arrhythmogenesis, fibrosis, apoptosis, hypertrophy and ventricular dysfunction [25,26,27]. Consistently, in adult cardiomyocytes [28,29], opioids activate δ and k receptors, which couple to Gi proteins and thus share part of their downstream signaling with adenosine and Bk. The δ receptor is most important in the conditioning phenomenon; not only does nonspecific opioid receptor blockade with naloxone abrogate protection by PC [30] and remote PC [31], but specific δ receptor blockade also largely abrogates protection by PC [32], postconditioning [33] and remote PC [34].

It has been demonstrated in an in vivo model of ischemia-reperfusion injury that, during PC, the concentration of Bk is rapidly increased [3], and it activates the BKR2 subtype, which couples to Gi proteins and activates the downstream endothelial nitric oxide synthase/protein kinase G and reperfusion injury salvage kinase pathways [35]. Bk also activates cyclo-oxygenase and prostacyclin synthesis to attenuate infarct size [1,2]. Moreover, the exogenous administration of Bk showed anti-apoptotic effects in rabbit hearts exposed to cardioplegic arrest [36] and reduced the percentage of senescence in endothelial cells exposed to oxidative stress [37]. In accord with these data, our findings demonstrate that PC evokes Bk synthesis and release in endothelial cells.

Our data show that Bk synthesis is due to increased enzymatic activity of KLK1 rather than augmented expression of KLK1 and Kn. Consistently, AP pretreatment to inhibit KLK1 activity abolished the PC-mediated cytoprotective effect. These results are in line with the evidence that, after PC application, KLK1-deficient mice showed a major extension in myocardial infarct size following ischemia-reperfusion injury with respect to wild-type mice [38], and that human KLK1 delivery into rats using an adenoviral vector, by increasing cardiac kinin levels, was associated with a significant limitation of the infarct size in the ischemic heart and the attenuation of apoptosis in the ischemic zone [39].

The results of our study show a prominent role of BKR2 in the induction of PC-mediated cytoprotection, subsequently supported by the increased expression of BKR1 in late PC. A large body of evidence confirms the key role of BKR2 in triggering the PC phenomenon. Yang et al. [40] showed that PC did not confer protection against infarction in BKR2 receptor knock-out mice. Furthermore, the inhibition of BKR2 by HOE-140 abrogated the reduction in infarct size mediated by late PC in rabbit hearts [41]. Less well known is the role of BKR1; nevertheless, there is limited evidence suggesting a role for BKR1 activation in PC. Using knockout mice missing the gene for either BKR1 or BKR2, a study by Duka et al. [42] showed that both BKRs contribute to the cardioprotective Bk-mediated effect of ACE inhibition. We found that the chemical inhibition and genetic silencing of BKR2 abolished the activation of both early and late PC-mediated cytoprotection, whereas the chemical inhibition and genetic silencing of BKR1 abrogated cell protection against prolonged hypoxia only in late PC, without exerting any effect in early PC.

Finally, our data show that PKA and Akt activation and their interaction during PC are abrogated by BKR inhibition, allowing the speculation that PC-induced Bk synthesis from bAECs promotes cytoprotection against prolonged hypoxia through this molecular pathway [12]. It is well known that Bk triggers PC in the heart through a pathway that includes Akt [43] and stimulates nitric oxide release from bAECs through a PKA-dependent mechanism [44]. Thus, our results show that PC promotes the activation of intracellular pathways leading to survival, not only as a consequence of neurohumoral effects, but also through the autocrine action of a molecular mediator (Bk) on cells. These findings highlight the translational value of autocrine hormonal stimulation in mediating molecular pathways in response to different triggers.

Endothelium is ubiquitous in the human body. Therefore, each molecule synthesized by endothelium might have an effect in all organs. We did not focus on the paracrine activity of Bk in our study, but our data seem to confirm the pivotal role of endothelium in the modulation of heart cell function [45]. In particular, they can further explain the greater positive effect of ACE-inhibitor therapy in the reduction of cardiovascular events in high risk-patients compared to angiotensin II receptor 1 blockers in clinical studies [46]. Our results are also consistent with the recent evidence that combined angiotensin receptor antagonism and neprilysin (an enzyme deputed to the degradation of natriuretic peptides and other vasoactive substances, including Bk) inhibition by sacubitril/valsartan has a pro-survival effect in reperfused rabbit hearts after myocardial infarction [47]. They may also explain the increased cardioprotective effect of combined remote ischemic conditioning and post-conditioning, both inducing Bk release in experimental settings, in conjunction with percutaneous coronary angioplasty (PCI), compared to PCI alone in ST elevation myocardial infarction patients [48].

In conclusion, a deeper understanding of the molecular pathways accounting for cytoprotection in endothelial cells could be useful to identify drugs that improve resistance to hypoxia in ischemic organs, like the heart, during acute myocardial infarction. Thus, our results may have important clinical relevance.

## 4. Materials and Methods

### 4.1. Cell Culture

BAECs were purchased from Lonza Biologics Inc. (Portsmouth, NH, USA); cultured in Dulbecco’s modified Eagle’s medium/Ham’s F-12 (DMEM; Lonza Biologics Inc.; Portsmouth, NH, USA) supplemented with 10% fetal bovine serum (FBS; Lonza Biologics Inc.; Portsmouth, NH, USA), 1% penicillin-streptomycin (Lonza Biologics Inc.; Portsmouth, NH, USA) and 1% glutamine (Lonza Biologics Inc.; Portsmouth, NH, USA); and maintained at 37 °C in 5% CO_2_. BAECs were used at passages 3–7.

### 4.2. Hypoxia

Hypoxia was induced by the incubation of cells in a medium previously saturated for 10 min at 1 atm with a 95% N_2_ and 5% CO_2_ mixture, containing 116 mM NaCl, 5.4 mM KCl, 0.8 mM MgSO_4_, 26.2 mM NaHCO_3_, 1 mM NaH_2_PO_4_, 1.8 mM CaCl_2_, 0.01 mM glycine and 0.001% (*w*/*v*) phenol red, and placement in an anaerobic chamber (hypoxia chamber) filled with the same gas mixture and heated to 37 °C. The pH, PO_2_ and PCO_2_ of the resulting medium were 7.36 ± 0.2, 45.3 ± 1.2 mmHg and 35.3 ± 0.8 mmHg, and 7.32 ± 0.9, 32.6 ± 1.1 and 37.9 ± 2.1 mmHg, before and at the end of hypoxia, respectively [49].

### 4.3. Estimation of Bradykinin (Bk) Production

We used an enzyme immunoassay (EIA) purchased from Kamiya Biomedical Company, according to manufacturer’s instructions.

Firstly, a Bk calibration semi-logarithmic curve was generated by progressive dilutions of a specific calibrator (Bk Calibrator; Kamiya Biomedical Company; 12779 Gateway Dr, Tukwila, WA 98168, USA) to quantify the Bk production obtained during the experiments.

Cells were plated on 100 mm dishes and serum-starved overnight. The next day, cells were stimulated as described and 500 μL of starvation medium (FBS-free) were collected from each well, transferred to a plastic tube and added with 100 μL of Deproteinizing Reagent. Then, this mixture was centrifuged at 3000 rpm for 10 min at 4 °C. Successively, 250 μL of the resulting supernatant were transferred into another plastic tube, added with 250 μL of Buffer Solution B (Human Bradykinin EIA Kit, Kamiya Biomedical Company; 12779 Gateway Dr, Tukwila, WA 98168, USA) and mixed to prepare the pretreatment sample.

The experiment wells and the reagents were allowed to come to room temperature before starting the assay. The next steps were as follows: (1) the desired number of Anti-Rabbit IgG-Coated Wells was removed for determination, 100 μL of prepared Bradykinin Antibody Solution were pipetted into these wells, and the solution was mixed with a plate rotator; after incubation at room temperature for 1 h, the reaction mixture was aspirated and the cells were washed with 300 μL of diluted Wash Buffer (Human Bradykinin EIA Kit, Kamiya Biomedical Company) with a plate washer (this procedure was repeated three times); then, the next procedure was immediately proceeded to without drying the wells; (2) 100 μL of Buffer Solution C (Human Bradykinin EIA Kit, Kamiya Biomedical Company) and 50 μL of each prepared Calibrator Solution (Human Bradykinin EIA Kit, Kamiya Biomedical Company) were added to predetermined wells (to be used for the calibration curve), 50 μL of Buffer solution A (Human Bradykinin EIA Kit, Kamiya Biomedical Company) and 100 μL of pretreatment sample were added to predetermined reaction wells, and all samples were mixed using a plate rotator, and incubated at room temperature for 1 h; (3) 50 μL of Bradykinin Enzyme Conjugate Solution (Human Bradykinin EIA Kit, Kamiya Biomedical Company) were added to each well, mixed with a plate rotator and incubated overnight at 4 °C; (4) the reaction mixture was removed with a plate washer and each well was washed with 300 μL of diluted Wash Buffer (Human Bradykinin EIA Kit, Kamiya Biomedical Company) (this procedure was repeated four times); after washing, the residue of the solution was removed by turning the plate upside down and tapping on a paper towel; (5) 100 μL of Substrate solution (Human Bradykinin EIA Kit, Kamiya Biomedical Company) were added to each well and allowed to incubate at room temperature for 30 min (during this step, experiment wells were protected from light by wrapping the plate with aluminum foil); (6) to stop the enzymatic reaction, 100 μL of Stop Solution were added (Human Bradykinin EIA Kit, Kamiya Biomedical Company); (7) all samples were stirred with a plate rotator; (8) the absorbance of each well was measured at 492 nm with a plate reader; (9) by using the calibration curve obtained, the Bk concentration corresponding to the absorbance of a pretreated sample was calculated (the Bk concentration is expressed as the weight in picograms per well of Bk in 100 mL of pretreated sample (unit: pg/well) to calculate the Bk concentration of the original sample; the reading values as expressed in pg/well were multiplied by 24 (a fixed coefficient, considering the molecular weight of Bk).

### 4.4. Reverse Transcriptase Polymerase Chain Reaction (RT-PCR)

The expression of tissue kallikrein (KLK1), kininogen (Kn), and BKR1 and 2 during PC was assessed by RT-PCR. Specific primers were designed using the Roche Assay Design Center (available online at http://www.universalprobelibrary.com).

Total RNA was isolated using TRIzol reagent (Invitrogen), and cDNA was synthesized by means of a Thermo-Script RT-PCR System (Invitrogen), following the manufacturer’s instructions. After reverse transcription, real-time quantitative PCR was performed with the SYBR Green real-time PCR master mix kit (Applied Biosystems) and quantified by built-in SYBR Green Analysis (Applied Biosystem) on a StepOne instrument (Applied Biosystem). The primers for gene analysis were as follows. KLK1: forward 5′-GAGCACCTGCCTGGCCTCCGGCTGGGGC-3′, reverse 5′-ACACATGAGGGGGCCCCCTGAGTCACCC-3′; Kn: forward 5′-TGACAGCCCAGTATGAGTGC-3′, reverse 5′-AGTCGGGGCTCTTGGTAGAT-3′; BKR1: forward 5′-GCCAACTTCTTTGCTTTCGT-3′, reverse 5′-TGGCCCCAGAAGACATAAAT-3′; BKR2: forward 5′-CTTCCTGGACACGCTGCT-3′, reverse 5′-TGTGTGAAGATGTCGATCACG-3′ and 18S: forward 5′-GTAACCCGTTGAACCCATT-3′, reverse 5′-CCATCCAATCGGTAGTAGCG-3′.

### 4.5. Measurement of Tissue Kallikrein Activity (KLK1) in Endothelial Cells

Confluent monolayers of bAECs in 100 mm dishes were serum-starved overnight. The next day, the cells were stimulated as described and washed twice with phosphate-buffered saline (PBS; Sigma Aldrich Corporation; St. Louis, MO; USA), scraped off the dish with a rubber policeman, suspended in 0.5 mL Tris buffer (0.1 mol/L, pH 7.4), containing 1 mL/L Tween 20 (Sigma Aldrich Corporation; St. Louis, MO; USA), then sonicated for 5 s (20% Output energy) with a Sonifier B-12 (Branson Sonic Power Co., Danbury, Co., USA) on ice. Samples of the cell homogenate were assayed immediately for KLK1 activity.

KLK1 activity was assessed by the colorimetric measurement of the release of 4-nitroaniline [15]. Briefly, 20 μL of Standard (0.1, 0.25, 0.5, 1, 2, 4, 8 mU porcine tissue kallikrein; Sigma Aldrich Corporation; St. Louis, MO, USA) or sample were diluted in a total volume of 150 μL (0.1 mol/L Tris, 0.234 mol/L NaCl, pH 7.8). The enzymatic reaction was started by the addition of 150 μL of 0.5 mmol/L Benzoyl-Pro-Phe-Arg-4-nitranilide acetate (Chromozym PK, Roche Diagnostics GmbH, Germany) in the presence and absence of 200 klU aprotinin (Sigma Aldrich). The absorbance at 405 nm (Titertek MCC 340; Lausanne, Switzerland) was measured to determine the 4-nitroaniline release after 1, 2, 3 and 4 min. KLK1 activity was calculated, comparing the mean values of change in absorbance/minute in standards and in samples. Only the activity of the reaction inhibited by aprotinin (AP) was attributed to KLK1-like activity.

### 4.6. Immunoblotting and Immunoprecipitation

Cells washed twice with ice-cold Ca^2+/^Mg^2+^ -free Dulbecco PBS and lysed with buffer containing 50 mM HEPES (pH 7.6), 1mM EDTA, 5 mM EGTA, 10 mM MgCl_2_, 50 mM β-glycerophosphate, 1 mM vanadate, 10 mM sodium fluoride, 30 mM sodiumpyrophosphate, 2 mM dithiothreitol, 1 mM AEBSF were used for the detection of Akt phosphorylation by anti-phospho-Akt serine473 antibody (Cell Signaling Technology). Cells lysed with buffer containing 150 mM NaCl, 50 mM Tris (pH 7.5), 0.5% deoxycholic acid, 1% NP-40 (IGEPAL CA-630), 0.1% SDS, 0.1 mM sodium orthovanadate, 1 mM sodium fluoride, 0.5 mM PMSF, 0.5 μg/mL aprotinin and 0.5 μg/mL leupeptin were used for other immunoblots and immunoprecipitations. Immunoblots and immunoprecipitates were subjected to SDS-PAGE, transferred to a polyvinylidene difluoride (PVDF) membrane and assessed with primary antibodies.

We used the following primary antibodies: anti-BKR1 and 2, anti-Gsα, anti-actin, anti-Akt, anti-phospho-Akt ser473, anti-PKA catalytic subunit, anti-KLK1 (Santa Cruz Technology) and anti-cleaved caspase-3 (Cell Signaling Technology).

Horseradish peroxidase-conjugated (Cell Signaling and Santa Cruz Technology) antibodies were used as secondary antibodies. The bound secondary antibodies were detected by enhanced chemiluminescence (Amersham Pharmacia Biotech).

### 4.7. Apoptosis and Necrosis Detection

Cells undergoing early apoptosis were stained with Annexin V (AV), but not with Propidium Iodide (PI) (Annexin-V- FLUOS Staining Kit, Roche), whereas necrotic cells or cells in late apoptosis stages were stained with both AV and PI. The percentage of apoptotic and necrotic cells was calculated by dividing the number of AV-positive/PI-negative cells and AV-positive/PI-positive cells by the total number of nuclei detected with DAPI staining, respectively [49].

### 4.8. Gene Silencing by Transient Transfection

BKR1 short interfering RNA (BKR1-siRNA) was generated from the bovine BKR1 sequence found 489 bp after the start codon (GCCCTCCTGAGTCTCCCCA; EUROGENTEC S.A., Belgium).

BKR2 short interfering RNA (BKR2-siRNA) was generated from the bovine BKR2 sequence found 509 bp after the start codon (CTTCCTCATGCTGGTGAGC; EUROGENTEC S.A., Belgium).

BAECs were plated at a density of 1 × 10^6^ per well in six-well plates. Twenty-four hours after plating, the medium was changed to DMEM without supplements. Transfections were carried out using 10 µL/mL of LipofectAMINE (Life Technologies) in 1 mL/well of Dulbecco’s modified Eagle’s medium. Four hours after transfection, the culture medium was changed to the culture medium supplemented with serum. Forty-eight hours after transfection, BAECs were serum-starved overnight and stimulated as described.

### 4.9. Preparation of Membrane and Cytosol Extracts

Cells were plated on 100 mm dishes and serum-starved overnight. The next day, the cells were stimulated as described, washed twice with ice-cold Ca^2+^/Mg^2+^-free Dulbecco PBS and scraped with 100 μL of homogenization buffer (TRIS 25 mM; EDTA 5 mM, pH 8; EGTA 5 mM; PMSF 1 mM; aprotinin 10 μg; leupeptin 10 μg). Then, cells were centrifuged at 2500 rpm for 15 minutes. The supernatant was then centrifuged at 15,000 rpm for 15 min. The new supernatant was saved as the cytosolic fraction. The pellet (membrane) was resuspended in 1 mL of purification buffer (75 mM TRIS-Cl, pH 7.4; 12.5 mM MgCl_2_; 2 mM EDTA; 1 mM PMSF; 10 μg aprotinin; 10 μg leupeptin) and then centrifuged at 15,000 rpm for 15 min. The supernatant was discarded and the pellet was resuspended in 30 μL of purification buffer. The entire procedure was carried out at 4 °C. The samples were stored at −85 °C.

### 4.10. PKA Activity Assay

The activity of PKA was tested in a cell-free reaction system using the synthetic peptide kemptide (Kp) as a specific substrate. The assay was conducted as previously described [50]. BAECs were lysed in a buffer containing 50 mM ß-glycerophosphate, 1 mM NaF, 1.5 mM EGTA, 1% NONIDET *p*-40, 1 mM EDTA, 0.1 mM PMSF, 10 µg/mL leupeptin, 10 µg/mL aprotinin and 1 mM DTT. Phosphorylation was carried out in a final volume of 50 µL containing 50 mM HEPES (pH 7.5), 10 mM MgCl_2_ and 1 µg Kp. Phosphorylation was initiated by the addition of 10 µM (γ-32P) ATP (6 Ci/mmol; Bcs Biotech SpA; Italy). The reaction was allowed to proceed for 20 min at room temperature, arrested by the addition of sample buffer (125 mM Tris, pH 6.8; 4% SDS; 10% glycerol; 0.006% bromophenol blue; 2% ß-mercaptoethanol) and boiled for 3 min at 95 °C. The samples were subjected to 10% SDS–PAGE and autoradiography. The phosphorylation of Kp by the catalytic subunit of PKA (Sigma Aldrich Corporation, St. Louis, MO; USA) served as a positive control.

### 4.11. Statistical Analysis

Data are given as mean ± SEM. Statistical analyses were performed using analysis of variance. The post-test comparison was performed by the method of Tukey. A chi-squared test was used for categorical variables. Significance was accepted at *p* < 0.05 levels.

## Figures and Tables

**Figure 1 ijms-21-02965-f001:**
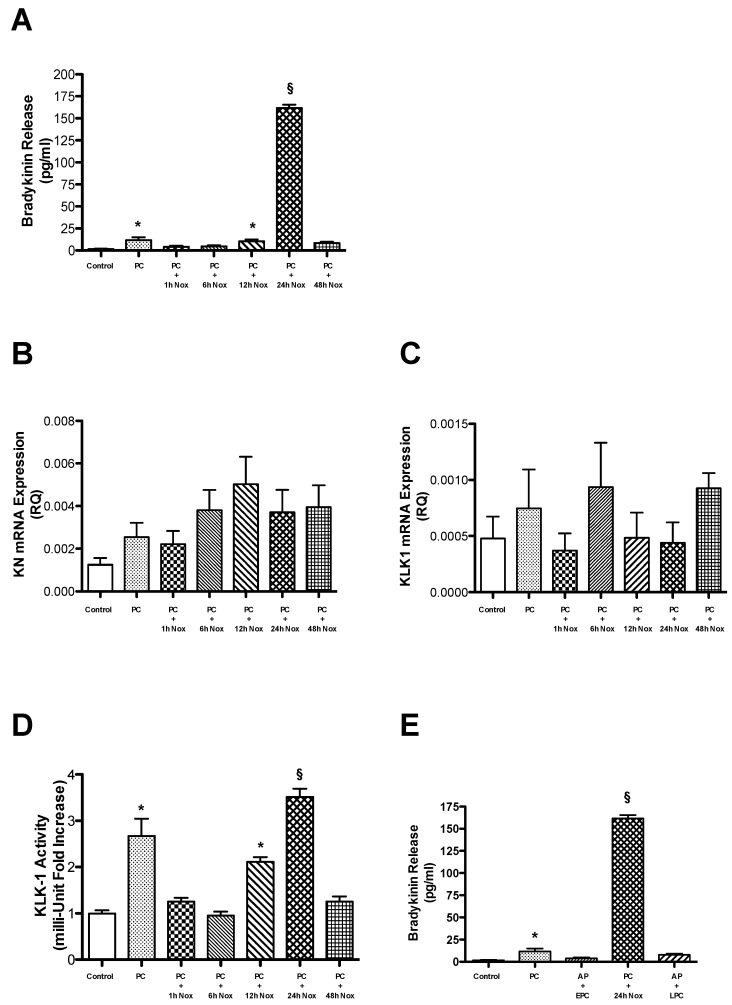
Cells were subjected to preconditioning (PC). (**A**) Bradykinin (Bk) production was assessed at different times following PC (Nox: normoxia). The bar graph shows the concentration (mean ± SEM) of Bk, representative of four independent experiments. The transcription of the mRNAs coding for (**B**) kininogen (Kn) and (**C**) tissue kallikrein (KLK1) was assessed at different times following PC. The bar graph represents the mean ± SEM, expressed as the RQ value, of four independent experiments. (**D**) KLK1 activity was assessed at different times following PC. The bar graph shows the mean ± SEM, expressed as the fold increase in KLK1 activity over that in control cells, of five independent experiments. (**E**) Bk synthesis was measured in early and late preconditioned cells, in the presence and absence of a KLK1 selective inhibitor, aprotinin (AP). The bar graph shows the concentration (mean ± SEM) of Bk, representative of three independent experiments. Nox: normoxia. * *p* < 0.001 vs. control; ^§^
*p* < 0.001 vs. control and PC, by one-way ANOVA with a post hoc test of HSD.

**Figure 2 ijms-21-02965-f002:**
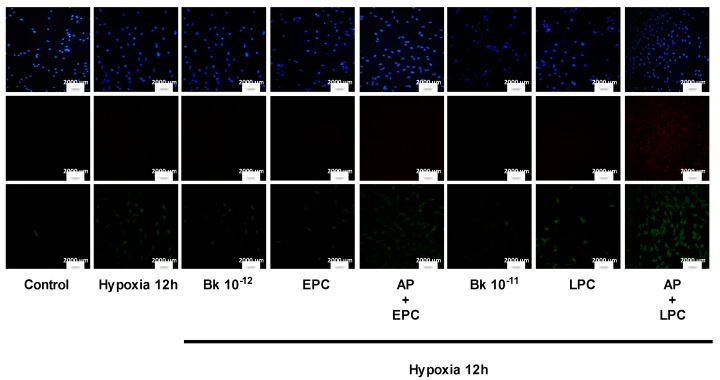
Cells were subjected to prolonged hypoxia (12 h) after early and late preconditioning (EPC and LPC), in the absence and presence of aprotinin (KLK1 selective inhibitor), and after exogenous bradykinin (Bk) administration (10^−12^ and 10^−11^ M), as described in the text. Apoptosis was assessed using Annexin V (green), and necrosis was assessed using Propidium Iodide (red) (PI) staining; nuclei were stained with DAPI (blue). The rates of apoptosis and necrosis were calculated by dividing the number of Annexin-V-positive/PI-negative cells and Annexin-V-positive/PI-positive cells, respectively, by the total number of nuclei detected with DAPI staining. The percentage of apoptotic and necrotic cell death was calculated from six independent experiments. Quantification of apoptosis and necrosis is reported in Appendix A. KLK1: tissue kallikrein. The size of the scale bar is 2000 µm.

**Figure 3 ijms-21-02965-f003:**
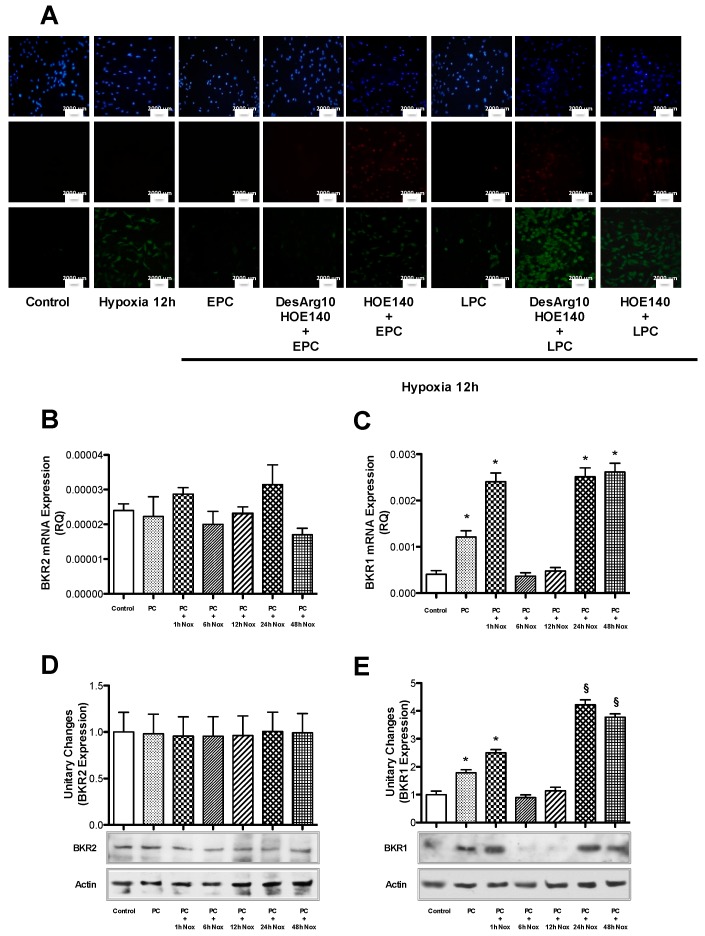
(**A**) Cells were subjected to prolonged hypoxia (12 h) after early and late preconditioning (EPC and LPC), in the absence and presence of selective bradykinin receptor 1 (BKR1) and 2 (BKR2) inhibitors (desArg10-HOE-140 and HOE-140, respectively). Cells were stained with Annexin-V (green), Propidium Iodide (PI, red) and DAPI (blue). The rates of apoptosis and necrosis were calculated by dividing the number of Annexin-V-positive/PI-negative cells and Annexin-V-positive/PI-positive cells, respectively, by the total number of nuclei detected with DAPI staining. The percentage of apoptotic and necrotic cell death was calculated from four independent experiments. Quantification of apoptosis and necrosis is reported in Appendix A. The size of the scale bar is 2000 µm. (**B**) The transcription of the mRNA coding for (**B**) BKR2 and (**C**) BKR1 was assessed at different times following PC (Nox: normoxia). The bar graph represents the mean ± SEM, expressed as the RQ value, from four independent experiments. The protein expression of BKR2 (**D**) and BKR1 (**E**) was assessed at different times following PC. The bar graph shows the mean ± SEM, expressed as the fold increase in BKR2 and BKR1 expression over that in control cells, of three independent experiments. Nox: normoxia. * *p* < 0.001 vs. control; ^§^
*p* < 0.001 vs. control and PC + 1 h Nox, by one-way ANOVA with a post hoc test of HSD.

**Figure 4 ijms-21-02965-f004:**
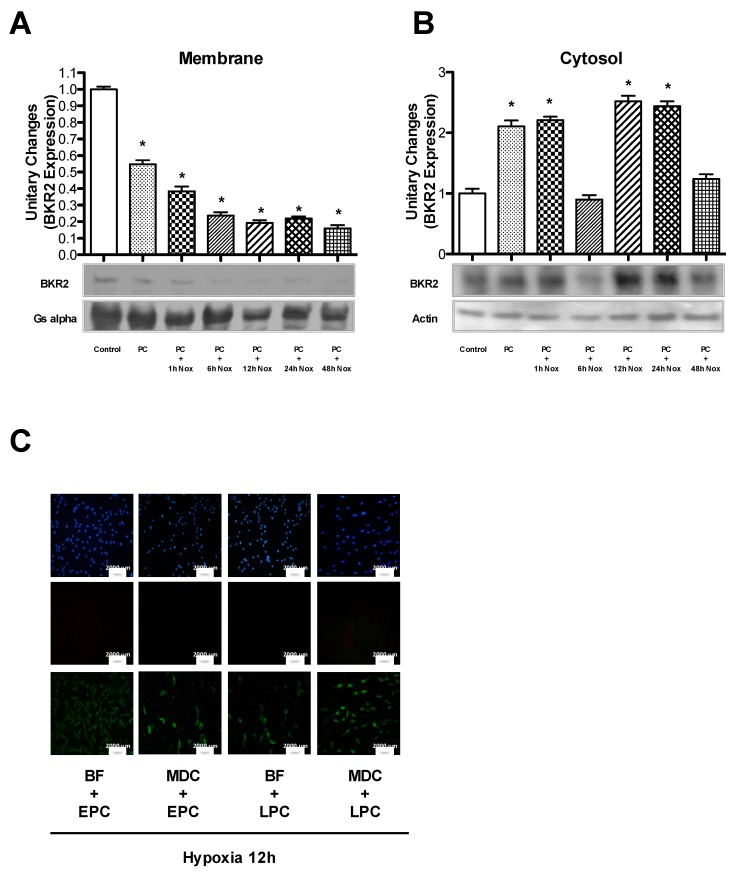
Cells were subjected to preconditioning (PC). Membrane (50 µg) (**A**) and cytosolic (50 µg) (**B**) fractions were subjected to immunoblot analyses using anti-BKR2 antibody. The bar graph represents densitometric analysis, with the mean ± SEM, expressed as fold increase/decrease in protein translocation over that in control cells, of three independent experiments. (**C**) Cells were subjected to prolonged hypoxia (12 h) after early and late preconditioning (EPC and LPC), in the absence and presence of selective clathrin (monodansylcadaverine) and lysosomal vacuolar ATPase (bafilomycin) inhibitors, as reported in the text. Cells were stained with Annexin-V (green), Propidium Iodide (PI, red) and DAPI. The rates of apoptosis and necrosis were calculated by dividing the number of Annexin-V-positive/PI-negative cells and Annexin-V-positive/PI-positive cells, respectively, by the total number of nuclei detected with DAPI staining. The percentage of apoptotic and necrotic cell death was calculated from four independent experiments. Quantification of apoptosis and necrosis is reported in Appendix A. The size of the scale bar is 2000 µm. Nox: normoxia; BKR2: bradykinin receptor 2; BF: bafilomycin; MDC: monodansylcadaverine. * *p* < 0.001 vs. control, by one-way ANOVA with a post hoc test of HSD.

**Figure 5 ijms-21-02965-f005:**
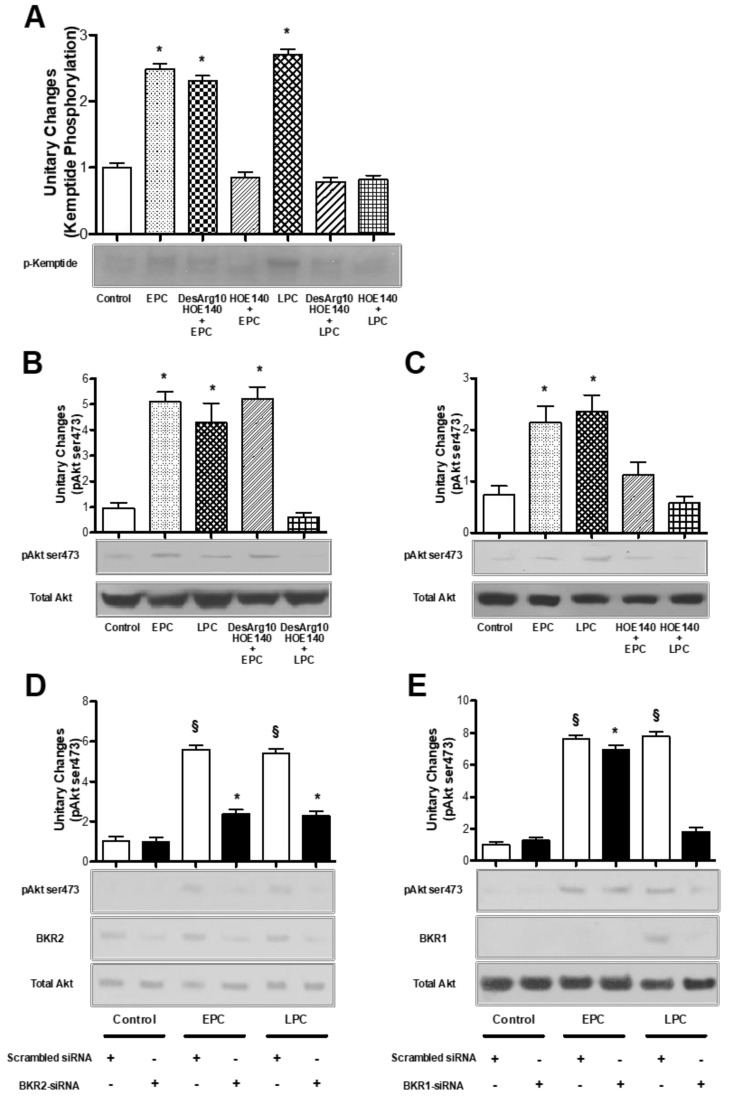
(**A**–**C**) Early and late preconditioned bAECs, in the absence and presence of selective bradykinin receptor 1 (BKR1) and 2 (BKR2) inhibitors (desArg10-HOE-140 and HOE-140, respectively), were harvested and lysed. The phosphorylation of kemptide (**A**) and Akt (**B**,**C**) was assessed. The bar graph represents densitometric analysis, with the mean ± SEM, expressed as fold increase in protein phosphorylation over that in control cells, of three independent experiments. (**D**,**E**) bAECs were transfected with bovine BKR2- and BKR1-short interfering RNA (BKR2-siRNA and BKR1-siRNA, 20 µg), then subjected to EPC and LPC. Akt phosphorylation at serine473 residues and BKR2 and BKR1 protein expression were assessed by immunoblotting. The bar graph represents densitometric analysis, with the mean ± SEM, expressed as fold increase in protein phosphorylation over that in control cells, of three independent experiments. EPC: early preconditioning; LPC: late preconditioning. * *p* < 0.001 vs. control; ^§^
*p* < 0.001 vs. control and siRNA transfection, by one-way ANOVA with a post hoc test of HSD.

**Figure 6 ijms-21-02965-f006:**
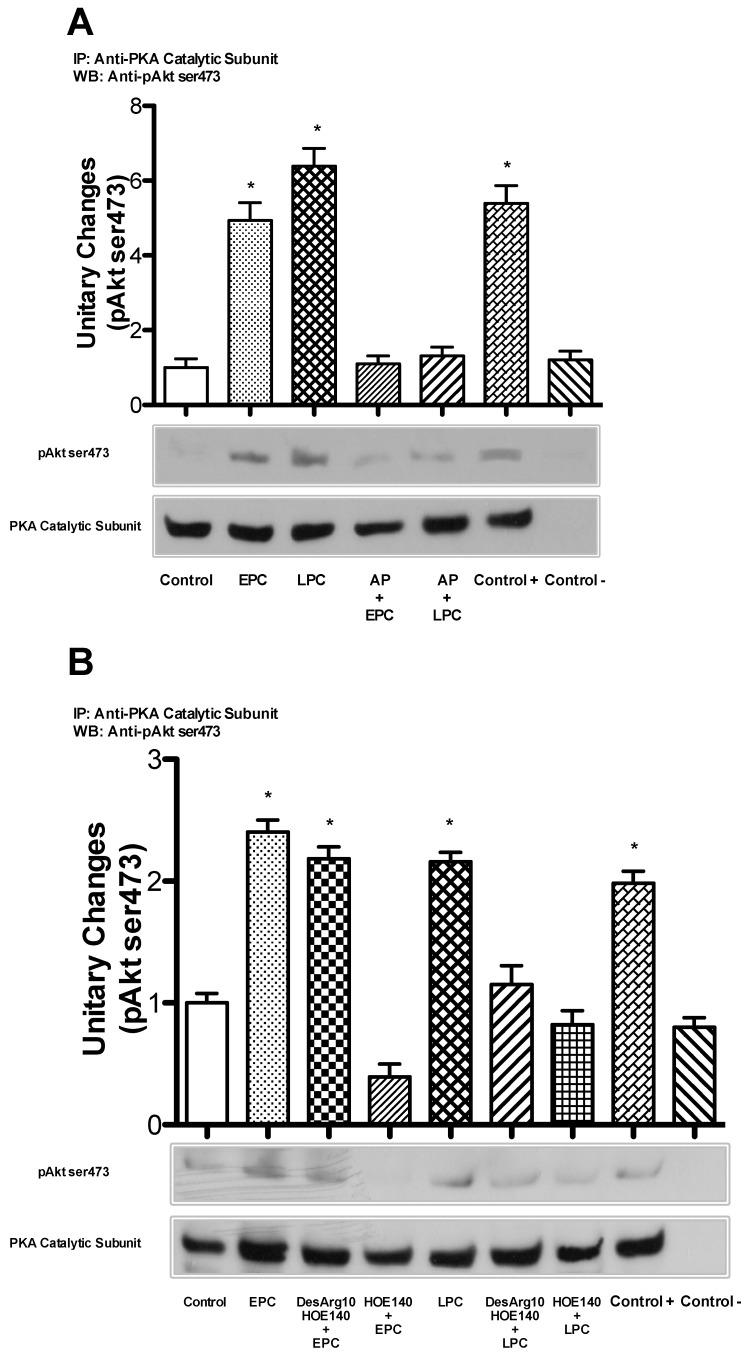
(**A**) Early and late preconditioned cells, in the absence and presence of aprotinin (KLK1 selective inhibitor), were harvested, immunoprecipitated with an antibody recognizing the catalytic subunit of protein kinase A (PKA) and immunoblotted with anti-phospho-Akt antibody. (**B**) Early and late preconditioned cells, in the absence and presence of bradykinin receptor 1 (BKR1) and 2 (BKR2) selective inhibitors (desArg10-HOE-140 and HOE-140, respectively), were harvested, immunoprecipitated with an antibody recognizing the catalytic subunit of PKA and immunoblotted with anti-phospho-Akt antibody. The bar graph represents densitometric analysis, with the mean ± SEM, expressed as fold increase in protein co-immunoprecipitation over that in control cells, of three independent experiments. AP: aprotinin; EPC: early preconditioning; LPC: late preconditioning; Control +: LPC lysate without beads; Control -: PKA catalytic subunit antibody plus beads without cell lysate. * *p* < 0.001 vs. control, by one-way ANOVA with a post hoc test of HSD.

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
