# Peer review of "Autocrine Bradykinin Release Promotes Ischemic Preconditioning-Induced Cytoprotection in Bovine Aortic Endothelial Cells"

_ijms, 2020, doi:10.3390/ijms21082965_

Round 1
Reviewer 1 Report
The work presented is well planned scientifically and illustrated graphically. The methodology is clear and the results presented convincing. However, it is not a discovery work. It presents the known problem of the bradykinin defense mechanism in the protection of cell ischemia many times already proven.
eg.
1.Bernier, S.G., S. Haldar, and T. Michel, Bradykinin-regulated interactions of the mitogen-activated protein kinase pathway with the endothelial nitric-oxide synthase. J Biol Chem, 2000. 275(39): p. 30707-15.
2. Jalowy A, Schulz R, Dorge H et al. Infarct size reduction by AT1-receptor blockade through a signal cascade of AT2-receptor activation, bradykinin and prostaglandins in pigs. Journal of the American College of Cardiology 1998; 32: 1787-96.
and many others.
Despite this, it seems that it can be a supplement to confirmation of knowledge
Author Response
Response to Reviewer 1 Comments
We thank Reviewer 1 for her/his constructive criticisms.
Point 1: The work presented is well planned scientifically and illustrated graphically. The methodology is clear and the results presented convincing. However, it is not a discovery work. It presents the known problem of the bradykinin defense mechanism in the protection of cell ischemia many times already proven. Despite this, it seems that it can be a supplement to confirmation of knowledge.
Response 1: We agree with the criticism raised by the Reviewer 1 that bradykinin (Bk) is well known to play a pivotal role as a systemic mediator of ischemic preconditioning (PC) [1-4]. We also remarked this concept in the introduction of our paper. However, so far, it has not been demonstrated that Bk is synthesized by endothelial cells, during PC. Furthermore, the principal novelty of our results is represented by the fact that Bk synthesis evoked by PC mediates the protection against both apoptotic and necrotic hypoxia-induced cell death in an autocrine manner.
In order to reinforce the key role of endothelium in autocrine and paracrine modulation of cells function, especially in the heart, we implemented the last part of discussion by adding some statements about clinical implications and pharmacological perspectives deriving from our data (page 11, from line 224 to line 238 of the revised version of the manuscript).
References:
- Jalowy A, Schulz R, Dorge H et al. Infarct size reduction by AT1-receptor blockade through a signal cascade of AT2-receptor activation, bradykinin and prostaglandins in pigs. Journal of the American College of Cardiology1998; 32: 1787-96.
- Penna C, Mancardi D, Tullio F, Pagliaro P. Postconditioning and intermittent bradykinin induced cardioprotection require cyclooxygenase activation and prostacyclin release during reperfusion. Basic research in cardiology2008; 103: 368-77.
- Schulz R, Post H, Vahlhaus C, Heusch G. Ischemic preconditioning in pigs: a graded phenomenon: its relation to adenosine and bradykinin. Circulation1998; 98: 1022-9.
- Sharma R, Randhawa PK, Singh N, Jaggi AS. Bradykinin in ischemic conditioning-induced tissue protection: Evidences and possible mechanisms. European journal of pharmacology2015; 768: 58-70.

We thank Reviewer 1 for her/his constructive criticisms.
Point 1: The work presented is well planned scientifically and illustrated graphically. The methodology is clear and the results presented convincing. However, it is not a discovery work. It presents the known problem of the bradykinin defense mechanism in the protection of cell ischemia many times already proven. Despite this, it seems that it can be a supplement to confirmation of knowledge.
Response 1: We agree with the criticism raised by the Reviewer 1 that bradykinin (Bk) is well known to play a pivotal role as a systemic mediator of ischemic preconditioning (PC) [1-4]. We also remarked this concept in the introduction of our paper. However, so far, it has not been demonstrated that Bk is synthesized by endothelial cells, during PC. Furthermore, the principal novelty of our results is represented by the fact that Bk synthesis evoked by PC mediates the protection against both apoptotic and necrotic hypoxia-induced cell death in an autocrine manner.
In order to reinforce the key role of endothelium in autocrine and paracrine modulation of cells function, especially in the heart, we implemented the last part of discussion by adding some statements about clinical implications and pharmacological perspectives deriving from our data (page 11, from line 224 to line 238 of the revised version of the manuscript).
References:
- Jalowy A, Schulz R, Dorge H et al. Infarct size reduction by AT1-receptor blockade through a signal cascade of AT2-receptor activation, bradykinin and prostaglandins in pigs. Journal of the American College of Cardiology 1998; 32: 1787-96.
- Penna C, Mancardi D, Tullio F, Pagliaro P. Postconditioning and intermittent bradykinin induced cardioprotection require cyclooxygenase activation and prostacyclin release during reperfusion. Basic research in cardiology 2008; 103: 368-77.
- Schulz R, Post H, Vahlhaus C, Heusch G. Ischemic preconditioning in pigs: a graded phenomenon: its relation to adenosine and bradykinin. Circulation 1998; 98: 1022-9.
- Sharma R, Randhawa PK, Singh N, Jaggi AS. Bradykinin in ischemic conditioning-induced tissue protection: Evidences and possible mechanisms. European journal of pharmacology 2015; 768: 58-70.
Reviewer 2 Report
In this manuscript titled, " Autocrine Bradykinin Release Promotes Ischemic Preconditioning-Induced Cytoprotection in Bovine Aortic Endothelial Cells", Alessandro Belliset al., authors investigate whether ischemic preconditioning (PC) induces bradykinin (Bk) synthesis in bovine aortic endothelial cells (bAECs) and relate the molecular mechanisms under this peptide provides cytoprotection against hypoxia. This manuscript appears preliminary.
- PC didn’t increase the mRNA level of KLK1 but an increase in the enzymatic activity of KLK1, authors should supply the WB data of the protein level of KLK1.
- The authors revealed that PC-mediated Bk against hypoxia-induced apoptosis. Is there any difference in protein levels of apoptosis-related Bax and Bcl-2?
Author Response
Response to Reviewer 2 Comments
We thank Reviewer 2 for her/his constructive criticisms.
Point 1: PC didn’t increase the mRNA level of KLK1 but an increase in the enzymatic activity of KLK1, authors should supply the WB data of the protein level of KLK1.
Response 1: As required, we added in supplementary material section Western blot analysis showing protein expression of tissue Kallikrein (KLK1) during ischemic preconditioning (PC). Consistently with mRNA levels, PC does neither increase protein expression of KLK1 (page 2 and Figure S1 of the revised version of supplementary material).
Point 2: The authors revealed that PC-mediated Bk protects against hypoxia-induced apoptosis. Is there any difference in protein levels of apoptosis-related Bax and Bcl-2?
Response 2: It has been yet demonstrated that greater increases of Bcl-xL and Bcl-2, but a lower level of increase of Bax, are observed in ischemic preconditioned heart [1, 2]and kidney [3]. We did not assess protein levels of apoptosis-related Bax and Bcl-2 in our experimental model. However, because Bad acts as an anti-apoptotic molecule when phosphorylated at ser112 and ser136 [4], we have previously assessed Bad phosphorylation in late preconditioned cells. In particular, we showed that late ischemic PC induced phosphorylation of both ser112 and ser136 [5]. Interestingly, these are the consensus sites for phosphorylation by PKA and Akt, respectively (Figure VI C in Data Supplement) [5].
References
- Min F, Jia XJ, Gao Q et al. Remote ischemic post-conditioning protects against myocardial ischemia/reperfusion injury by inhibiting the Rho-kinase signaling pathway. Experimental and therapeutic medicine2020; 19: 99-106.
- You L, Pan YY, An MY et al. The Cardioprotective Effects of Remote Ischemic Conditioning in a Rat Model of Acute Myocardial Infarction. Medical science monitor : international medical journal of experimental and clinical research2019; 25: 1769-79.
- Jang HS, Kim J, Kim KY et al. Previous ischemia and reperfusion injury results in resistance of the kidney against subsequent ischemia and reperfusion insult in mice; a role for the Akt signal pathway. Nephrology, dialysis, transplantation : official publication of the European Dialysis and Transplant Association - European Renal Association2012; 27: 3762-70.
- Downward J. How BAD phosphorylation is good for survival. Nature cell biology1999; 1: E33-5.
- Bellis A, Castaldo D, Trimarco V et al. Cross-talk between PKA and Akt protects endothelial cells from apoptosis in the late ischemic preconditioning. Arteriosclerosis, thrombosis, and vascular biology2009; 29: 1207-12.

We thank Reviewer 2 for her/his constructive criticisms.
Point 1: PC didn’t increase the mRNA level of KLK1 but an increase in the enzymatic activity of KLK1, authors should supply the WB data of the protein level of KLK1.
Response 1: As required, we added in supplementary material section Western blot analysis showing protein expression of tissue Kallikrein (KLK1) during ischemic preconditioning (PC). Consistently with mRNA levels, PC does neither increase protein expression of KLK1 (page 2 and Figure S1 of the revised version of supplementary material).
Point 2: The authors revealed that PC-mediated Bk protects against hypoxia-induced apoptosis. Is there any difference in protein levels of apoptosis-related Bax and Bcl-2?
Response 2: It has been yet demonstrated that greater increases of Bcl-xL and Bcl-2, but a lower level of increase of Bax, are observed in ischemic preconditioned heart [1, 2] and kidney [3]. We did not assess protein levels of apoptosis-related Bax and Bcl-2 in our experimental model. However, because Bad acts as an anti-apoptotic molecule when phosphorylated at ser112 and ser136 [4], we have previously assessed Bad phosphorylation in late preconditioned cells. In particular, we showed that late ischemic PC induced phosphorylation of both ser112 and ser136 [5]. Interestingly, these are the consensus sites for phosphorylation by PKA and Akt, respectively (Figure VI C in Data Supplement) [5].
References
- Min F, Jia XJ, Gao Q et al. Remote ischemic post-conditioning protects against myocardial ischemia/reperfusion injury by inhibiting the Rho-kinase signaling pathway. Experimental and therapeutic medicine 2020; 19: 99-106.
- You L, Pan YY, An MY et al. The Cardioprotective Effects of Remote Ischemic Conditioning in a Rat Model of Acute Myocardial Infarction. Medical science monitor : international medical journal of experimental and clinical research 2019; 25: 1769-79.
- Jang HS, Kim J, Kim KY et al. Previous ischemia and reperfusion injury results in resistance of the kidney against subsequent ischemia and reperfusion insult in mice; a role for the Akt signal pathway. Nephrology, dialysis, transplantation : official publication of the European Dialysis and Transplant Association - European Renal Association 2012; 27: 3762-70.
- Downward J. How BAD phosphorylation is good for survival. Nature cell biology 1999; 1: E33-5.
- Bellis A, Castaldo D, Trimarco V et al. Cross-talk between PKA and Akt protects endothelial cells from apoptosis in the late ischemic preconditioning. Arteriosclerosis, thrombosis, and vascular biology 2009; 29: 1207-12.